# Oxidative Stress-Induced Misfolding and Inclusion Formation of Nrf2 and Keap1

**DOI:** 10.3390/antiox11020243

**Published:** 2022-01-27

**Authors:** Vy Ngo, Nadun C. Karunatilleke, Anne Brickenden, Wing-Yiu Choy, Martin L. Duennwald

**Affiliations:** 1Department of Pathology and Laboratory Medicine, Schulich School of Medicine and Dentistry, University of Western Ontario, London, ON N6A 3K7, Canada; vngo23@uwo.ca; 2Department of Biochemistry, Schulich School of Medicine and Dentistry, University of Western Ontario, London, ON N6A 3K7, Canada; nkarunat@uwo.ca (N.C.K.); abricken@uwo.ca (A.B.); jchoy4@uwo.ca (W.-Y.C.); 3Department of Anatomy and Cell Biology, Schulich School of Medicine and Dentistry, University of Western Ontario, London, ON N6A 3K7, Canada

**Keywords:** Nrf2, Keap1, oxidation, oxidative stress, protein misfolding

## Abstract

Cells that experience high levels of oxidative stress respond by inducing antioxidant proteins through activation of the protein transcription factor nuclear factor erythroid 2-related factor 2 (Nrf2). Nrf2 is negatively regulated by the E3 ubiquitin ligase Kelch-like ECH-associated protein 1 (Keap1), which binds to Nrf2 to facilitate its ubiquitination and ensuing proteasomal degradation under basal conditions. Here, we studied protein folding and misfolding in Nrf2 and Keap1 in yeast, mammalian cells, and purified proteins under oxidative stress conditions. Both Nrf2 and Keap1 are susceptible to protein misfolding and inclusion formation upon oxidative stress. We propose that the intrinsically disordered regions within Nrf2 and the high cysteine content of Keap1 contribute to their oxidation and the ensuing misfolding. Our work reveals previously unexplored aspects of Nrf2 and Keap1 regulation and/or dysregulation by oxidation-induced protein misfolding.

## 1. Introduction

Oxidative stress is regulated by the transcription factor nuclear factor erythroid 2-related factor 2 (Nrf2) [1]. Nrf2 regulates the expression of a multitude of antioxidant genes and is negatively regulated by Kelch-like ECH-associated protein 1 (Keap1) [2], a substrate adaptor protein that binds to Nrf2 in the cytoplasm to promote its ubiquitination and ensuing degradation by the proteasome [2,3,4,5,6]. During oxidative stress, specific stress-sensing cysteine residues in Keap1 are oxidized [7,8,9], resulting in a conformational change in Keap1 that leads to dissociation of the Keap1-Nrf2 complex. This, in turn, leads to Nrf2 stabilization, nuclear translocation, and ultimately, activation of cytoprotective antioxidant genes [10,11]. Induction of the Keap1-Nrf2 antioxidant pathway is fundamental to protecting cells against oxidative stress, and mutations that disrupt Keap1-Nrf2 binding compromise Nrf2 regulation and can contribute to numerous human diseases [12,13]. The transcriptional activity of Nrf2 is therefore tightly regulated by its interaction with Keap1 [14,15]. Although Nrf2 is primarily regulated through its protein interactions, other factors may influence Nrf2 activity, such as competitive binding to the antioxidant response element (ARE) by transcriptional repressors (e.g., Bach1 [16]), phosphorylation of Nrf2 (e.g., GSK3, PKC [17,18,19]), and the intracellular re-localization of Nrf2 (e.g., PGAM5 [20]), among others. Given that Nrf2 is the key regulator of oxidative stress, the extent of oxidation within the Nrf2 protein itself may also influence its activity in the cell.

Nrf2 contains seven conserved regions that are referred to as the Nrf2-ECH homology (Neh) domains. Notably, the Neh2 domain that mediates Nrf2′s binding with Keap1 is mostly intrinsically disordered [14], and transiently structured elements within the Neh2, Neh7, and Neh1 domains of Nrf2 render the protein partially intrinsically disordered [21]. Intrinsically disordered proteins (IDPs) or proteins that contain long intrinsically disordered regions (IDRs) lack a fixed three-dimensional structure and are susceptible to protein misfolding and inclusion formation in cells due to their structural heterogeneity and flexible nature; however, this also allows for an enhanced binding capacity and multifunctionality [22,23,24]. This may explain Nrf2′s ability to bind to a vast array of different proteins. We use the term ‘protein misfolding’ here to indicate proteins that have acquired a non-native aberrant conformation, often in the form of inclusions or aggregates. Misfolded proteins can lose their normal function (i.e., loss of protein function) and tend to aggregate and form inclusions that can have deleterious effects on the cell (i.e., a toxic gain of function) [25]. Examples of disease-associated IDPs include α-synuclein in Parkinson’s disease [26] while in cancer, many key tumor suppressors contain long IDRs, such as p53 [27]. Interestingly, oxidative stress can affect the structural flexibility of IDPs/IDRs [26] and thus, may also modulate the folding or contribute to the misfolding of intrinsically disordered Nrf2, which may impair or alter its interactions with other proteins and thus, its transcriptional activity.

Protein misfolding may also expose hydrophobic or oxidation-prone cysteine residues to the surface of the protein, rendering them targets for oxidation by reactive oxygen species (ROS) and other oxidants [28]. Oxidation products of cysteines include disulfide bonds and mixed disulfide bonds. Oxidation can also lead to alteration of the non-covalent interactions within proteins, fragmentation of peptide chains, cross-linking of proteins, and/or oxidation of specific side chains, ultimately leading to protein destabilization and misfolding [29,30,31]. Cysteine residues are particularly susceptible to aberrant oxidation by ROS due to the presence of their nucleophilic thiol groups [32]. Keap1, which contains an exceptionally high percentage of cysteine residues, may thus be particularly susceptible to oxidation and inclusion formation.

In this study, we examined two aspects of protein oxidation: the misfolding of the intrinsically disordered Nrf2, and the misfolding of the cysteine-rich Keap1, under oxidative stress conditions. Using yeast, cultured mammalian cells, and purified proteins, we found that both Nrf2 and Keap1 misfold and form aberrant cytoplasmic and possibly nuclear protein inclusions upon exposure to high levels of oxidative stress. Our results suggest a previously unexplored mechanism by which the Keap1–Nrf2 interaction may be altered by oxidative stress as it pertains to protein misfolding and inclusion formation.

## 2. Materials and Methods

### 2.1. Prediction of Intrinsically Disordered Regions

Three independent algorithms, PrDOS, IUPred2A, and PONDR [33,34,35], were used to predict intrinsically disordered regions using protein sequences obtained from the UniProt [36] protein database. Using the scores obtained from each algorithm, each amino acid residue within the protein was assigned a numerical value of ‘+1′ or ‘−1′, where >0.5 = +1 (disordered) and values <0.5 = −1 (ordered). The scores were summed for each residue. Residues with a combined score of +3 across all three algorithms were considered disordered.

### 2.2. Protein Sequence Alignment

MEGA X [37] was used to perform protein sequence alignments using protein sequences obtained from the UniProt [36] protein database.

### 2.3. Yeast Growth Assays and Microscopy

For assessment of the relative growth, wild-type yeast and deletion strains obtained from the *Saccharomyces* Genome Deletion Project [38] were used. Yeast cells were transformed using the standard lithium acetate/salmon sperm carrier DNA/PEG method for the incorporation of yeast plasmids [39]. Transformed yeast cells were grown overnight in synthetic selective media to maintain these plasmids. Growth assays were performed by spotting 5× serial dilutions of OD_600_ = 0.2 on agar plates followed by incubation at 30 °C. Plates were imaged using the Bio-Rad ChemiDoc (Bio-Rad Laboratories, Hercules, CA, USA). Growth was quantified as previously described [40]. To assess protein expression using fluorescence microscopy, yeast was transformed with yellow fluorescent protein (YFP)-tagged plasmids. Cells were transferred to a glass microscope slide and coverslip and imaged using the Olympus BX-51 Bright Field/Fluorescence Microscope at 60× and captured using an equipped SPOT Pursuit CCD camera (SPOT Imaging, Sterling Heights, MI, USA).

### 2.4. Cell Lines and Culture Conditions

The HeLa cell line (human cervical cancer cell line) was maintained in Dulbecco’s Modified Eagle Medium (DMEM; Gibco Laboratories, Gaithersburg, MD, USA; 41966-029), supplemented with 10% FBS (Wisent Inc, Saint-Jean-Baptiste, QC, Canada; 080-150) and 1× penicillin-streptomycin (Corning Inc., Corning, NY, USA; 30-001-CI). Cells were cultured at 37 °C with 5% CO_2_. For transfections, cells were seeded in a 6-well plate at 1.0 × 10^6^ cells per well and grown to approximately 80% confluency. Cells were transfected with Lipofectamine LTX with PLUS Reagent (Thermo Fisher Scientific, Waltham, MA, USA; A12621) according to the manufacturer’s protocol in Opti-MEM I Reduced Serum Medium (Gibco Laboratories, Gaithersburg, MD, USA; 31985-062). Transfected cells were incubated at 37 °C for 6 h, followed by a wash in 1× PBS, and incubated in DMEM for 18 h at 37 °C. Cells were then split into the appropriate plates for subsequent experiments.

### 2.5. Cell Viability Assays

To determine the optimal hydrogen peroxide treatment concentration and duration, transfected HeLa cells were seeded into 96-well solid white microplates (Greiner Bio-One, Frickenhausen, Germany; M0187-32EA) at 4 × 10^4^ cells per well and incubated for 16 h. Cells were treated with H_2_O_2_ at the following concentrations (µM): 0, 100, 300, 500, 750, 1000, 1500, and 2000. Cell viability was assessed after 1, 3, 5, and 10 h using the CellTiter-Glo 2.0 Cell Viability Assay (Promega, Madison, WI, USA; G9242) according to the manufacturer’s protocol. Luminescence was measured using the Cytation 5 Cell Imaging Multi-Mode Reader (BioTek, Winooski, VT, USA).

### 2.6. Fluorescence and Immunofluorescence Microscopy

Transfected HeLa cells were seeded on a 15 mm circular glass coverslip (Matsunami Glass USA, Bellingham, WA, USA; C015001) in a 12-well plate at 1 × 10^5^ to ensure approximately 75% confluency the following day. Cells were treated with 300 µM H_2_O_2_ for 3 h. Cells were then fixed with 4% paraformaldehyde, permeabilized with 0.1% Triton X-100 in PBS, blocked with 20% goat head serum in PBB (0.5% BSA in PBS), and incubated with one of the following primary antibodies overnight at 4 °C at a concentration of 1:100: mouse anti-Nrf2 (Abcam, Cambridge, UK; ab62352) or mouse anti-Keap1 (Proteintech Group, Inc., Rosemont, IL, USA; 10503-2-AP). The coverslips were incubated with the following Alexa Fluor 488-conjugated secondary antibody for 1 h at room temperature at a concentration of 1:1500: goat anti-mouse (Thermo Fisher Scientific, Waltham, MA, USA; A-11094). Coverslips were mounted onto glass microscope slides with SlowFade Gold Antifade Reagent with DAPI (Thermo Fisher Scientific, Waltham, MA, USA; S36938) and cured at room temperature for 24 h. Cells were imaged using the Cytation 5 Cell Imaging Multi-Mode Reader (BioTek) using a 20× objective lens, or the Nikon Ti2-E Inverted Microscope paired with the A1R HD laser-scanning confocal (Nikon) using a 20× or 40× objective lens.

Protein inclusions were manually counted, and the following criteria were used to define a cell with inclusions: (1) to be included in the cell count, a cell’s DAPI-stained nucleus must be fully visible in the image (i.e., the nucleus cannot be cut off by the edge of the image); otherwise, that cell is excluded from the total cell number count; and (2) a cell must contain one or more distinguishable fluorescent foci to be defined as having inclusion(s). Following this criterion, the total number of cells with inclusions was divided by the total number of cells in the image to calculate the percentage of cells with inclusions. Three replicates per slide across three technical replicates were utilized.

### 2.7. RNA Isolation and Quantitative Reverse Transcription PCR (RT-qPCR)

HeLa cells were seeded in a 6-well plate at 1 × 10^6^ cells per well and grown to approximately 85% confluency the following day. Cells were treated with 300 µM H_2_O_2_ for 3 h. RNA was isolated using the RNA Isolation Kit (Advantech Co., Ltd., Taipei City, Taipei, Taiwan; AD100-31701), and 500 ng of RNA were converted into cDNA using the RevertAid H Minus First Strand cDNA Synthesis Kit (Thermo Fisher Scientific, Waltham, MA, USA; K1632). The PowerUp SYBR Green Master Mix (Applied Biosystems, Waltham, MA, USA; A25742) was utilized for quantitative PCR with the primer sequences listed in Table 1. The output values were normalized to *RPLP0* expression.

### 2.8. Protein Expression and Purification

Full-length wild-type Nrf2 and Nrf2 Neh5 were purified as described in our previous work [21]. The Keap1 Kelch domain was purified as described in [41]. Full-length Keap1 was purified as follows: Keap1 cloned into a pET1b plasmid was transformed into competent *E. coli* (Rosetta 2(DE3) pLysS) (Novagen, EMD Biosciences Inc., San Diego, CA, USA) by the calcium chloride method [42]. The cell culture was incubated in lysogeny broth (10 g/L tryptone, 5 g/L yeast extract, 10 g/L NaCl) at 37 °C until the OD_600_ reached ~0.7. Protein overexpression was induced with 1 mM isopropyl-β-D-thiogalactopyranoside (IPTG; Thermo Fisher Scientific, Waltham, MA, USA; R0392) at 18 °C for 21 h. Cultures were harvested by centrifugation at 6000 rpm for 15 min at 4 °C. Cell pellets were washed and resuspended in lysis buffer (50 mM Tris-HCl, pH 8.0, 150 mM NaCl, 5 mM MgSO_4_, 5 mM β-mercaptoethanol). The cell suspension was incubated with human lysozyme powder (10 mg/10 mL) for 30 min at 37 °C and pH was adjusted to 8.0 with 0.4 M Tris. The sample was homogenized using an Avestin EmulsiFlex C5 homogenizer (ATA Scientific, Caringbah, Australia). A SigmaFast Protease Inhibitor Cocktail tablet, EDTA-free (Sigma-Aldrich, St. Louis, MO, USA; S8830), 2 M NEM (250 µL/10 mL), and 200 mM PMSF (100 µL/10 mL) were added to the sample. The final concentrations of imidazole (1 M) and NaCl (5 M), and pH (~7.8–8.0) were adjusted. The sample was centrifuged at 40,000× *g* for 30 min at 4 °C. The supernatant was collected, and pH was adjusted to ~7.8 with HCl. The sample was resuspended with equilibrated Ni-Sepharose 6 Fast Flow beads (GE Healthcare, Chicago, IL, USA) and incubated for 2 h at room temperature on an orbital shaker. After 2 h, the slurry was poured into a collection column and washed with 100 mL of binding buffer (50 mM Tris-HCl, pH 8.0, 10 mM imidazole, 500 mM NaCl), followed by 200 mL of wash buffer (50 mM Tris-HCl, pH 8.0, 500 mM NaCl, 20 mM imidazole). The protein was eluted using 5 mL fractions of elution buffer (750 mM imidazole, 500 mM NaCl, 50 mM Tris-HCl, pH 8.0) and eluted fractions were monitored using the Bradford assay (Bio-Rad Laboratories, Hercules, CA, USA; 5000201). Fractions containing Keap1 protein were pooled up to a 20–25 mL total elution volume and dialyzed overnight at 4 °C with dialysis buffer (15 mM Tris-HCl, pH 8.0, 150 mM NaCl, 5 mM β-mercaptoethanol). The final protein concentration was determined by the Lowry assay (Thermo Fisher Scientific, Waltham, MA, USA; 23240). Using this method, we obtained 0.252 g of full-length Keap1 from 2 L of lysogeny broth (LB) growth media.

### 2.9. SDS-PAGE and Coomassie Blue Gel Staining

Purified protein (10 µg) was resolved on a 10% SDS-PAGE gel. The gel was stained with Coomassie Brilliant Blue (0.1% Coomassie Brilliant Blue R-250, 50% methanol (*v/v*), 10% glacial acetic acid (*v/v*), 40% H_2_O) for 30 min and de-stained overnight using a destaining solution (50% methanol (*v/v*), 10% glacial acetic acid (*v/v*), 40% H_2_O) with gentle agitation. Blots were imaged using the ChemiDoc Imaging System (Bio-Rad Laboratories, Hercules, CA, USA).

### 2.10. SDD-AGE (Semi-Denaturing Detergent Agarose Gel Electrophoresis)

Purified protein (10 µg) was resolved on a 1.8% agarose-2% SDS gel and run at 80 V for approximately 1.5 h at room temperature in 1× TAE-0.1% SDS running buffer. The gel was transferred at room temperature to PVDF by an overnight wet transfer by gravity according to the manufacturer’s protocol (Whatman TurboBlotter Transfer System, Cytiva, Marlborough, MA, USA). The membrane was blocked with 5% skim milk in PBST (1X PBS, 1% Tween-20) and incubated with one of the following primary antibodies overnight at 4 °C at a concentration of 1:1000: mouse anti-Nrf2 (Abcam, Cambridge, UK; ab62352) or mouse anti-Keap1 (Proteintech Group, Inc, Rosemont, IL, USA;10503-2-AP). The membrane was incubated with the following Alexa Fluor 680-conjugated antibody for 1 h at room temperature at a concentration of 1:1500: goat anti-mouse (Thermo Fisher Scientific, Waltham, MA, USA; A-21057). Blots were imaged using the ChemiDoc Imaging System (Bio-Rad Laboratories, Hercules, CA, USA).

### 2.11. Combined SDD-AGE and Fractionation Assay

For partitioning into soluble and insoluble fractions, purified protein was first aliquoted into a chilled microcentrifuge tube, which represents the ‘total’ fraction. A second aliquot was centrifuged at 10,000× *g* for 10 min at 4 °C. The supernatant was transferred into a chilled microcentrifuge tube, which represented the soluble ‘supernatant’ fraction. The pellet was resuspended in dialysis buffer (from the preceding protein purification process) and represented the insoluble ‘pellet’ fraction. Equal volumes of each fraction equivalent to 10 µg of the total fraction were resolved by SDD-AGE (see above).

### 2.12. Statistical Analysis

Statistical analyses were conducted using Prism 8 (GraphPad Software, San Diego, CA, USA). Statistical significance was obtained by performing a one-way ANOVA with Tukey post hoc for comparison between groups. Error bars represent the standard deviation. *p*-values less than 0.05 were considered statistically significant. Significance levels are indicated using asterisks, where * is *p* < 0.05, ** is *p* < 0.01, and *** is *p* < 0.001; *p* = ns is non-significant. Shapiro–Wilk tests were performed for all data sets to ensure normality.

## 3. Results

### 3.1. Analysis of Nrf2

#### 3.1.1. Nrf2 Is Intrinsically Disordered and Keap1′s High Cysteine Content Is Evolutionarily Conserved

Figure 1A schematically illustrates the functional domains of human Nrf2 and Keap1. Nrf2 contains 7 conserved Neh domains and 6 cysteine residues, whereas Keap1 contains 3 functional domains and a total of 27 cysteine residues. The key oxidative stress-sensing sensor cysteines in Keap1 are indicated with an asterisk (*) [7,8,9,43].

The disordered profile plots for Nrf2 and Keap1 indicate the location of intrinsically disordered regions as predicted by three independent algorithms (PrDOS, IUPred2A, and PONDR) [33,34,35] (Figure 1B) (refer to Appendix A for the individual algorithm predictions). Intrinsically disordered regions are highlighted in yellow. In our analyses, an amino acid residue was denoted as disordered if it was predicted as disordered by all 3 individual algorithms, i.e., a prediction value of >0.5 = +1 (disorder) and prediction value of <0.5 = −1 (order); thus, a combined value of +3 was denoted as disordered. From these results, Nrf2 was predicted to contain 11 intrinsically disordered regions with an overall combined percent disordered score of 39.34%. Our previous work also provides experimental evidence for the highly intrinsically disordered properties of Nrf2 [21]. In comparison, Keap1 is mostly ordered with a single predicted disordered region and an overall combined percent disordered score of 0.48%, which is also corroborated by structural studies of the folded Kelch domain of Keap1 [44].

Next, we analyzed the amino acid composition of Nrf2 and Keap1 and calculated the percentage of cysteine content across 15 metazoan species (Appendix A) from humans to zebrafish (Figure 1C). The total cysteine content in human Nrf2 was 0.99%, which is below the average for the human proteome of 2.3% [45], while the average for Nrf2 across all 15 species analyzed here was 1.11%. Intriguingly, human Keap1 contains a high 4.33% cysteine content, which is almost double the human average. The overall average for Keap1 across all species was 4.07%, which demonstrates that the high cysteine content in Keap1 is conserved, even beyond the key sensor cysteines that directly regulate Nrf2 interaction. To determine if cysteine residues are evolutionarily conserved regarding their positions within Keap1 and Nrf2 across the 15 species, we performed a protein sequence alignment using MEGA X [37] and found that all 6 cysteine residues in Nrf2 (highlighted in yellow) were either completely or highly conserved across species (i.e., perfectly aligned in 13–15 species) (Figure 1D, top). For Keap1, 24 of the 27 cysteine residues in human Keap1 were either completely or highly conserved (i.e., perfectly aligned in 12 species) (Figure 1D, bottom). Expectedly, all sensor cysteines within Keap1 (indicated with an asterisk (*)) were also either completely or almost completely conserved (i.e., perfectly aligned in 14 species).

#### 3.1.2. Oxidative Stress and Nrf2 and Keap1 Expression in Yeast

We previously established yeast as a useful tool to study Nrf2 interactions (Ngo et al., 2022; in press [46]) and re-capitulated our data showing that Nrf2 is toxic in yeast. Here, we again used growth assays to assess if Nrf2 expression in yeast is affected by the absence of oxidative stress genes. Human Nrf2 expressed in yeast caused ‘toxicity’, defined as an impaired growth phenotype on growth media compared to the empty vector control. Nrf2 was expressed in wild-type yeast and yeast strains deleted for an array of oxidative stress genes. Growth is quantified to the right as done previously [40] (Figure 2A). Only significant data is shown; for the complete list of deletion strains, refer to Appendix A. Nrf2 toxicity was not significantly altered in these deletion strains. While human Nrf2 is toxic in yeast, it is important to note that Nrf2 is not toxic in mammalian cells; as expected, Nrf2 promotes increased cell viability (determined by the quantification of ATP levels, which indicates the presence of metabolically active cells [47]) compared to the untreated control (Figure 2B). This antioxidant survival advantage is diminished in two mutant variants of Nrf2, L30F and T80R, which have a reduced or impaired capacity to interact with Keap1 [13]. On the other hand, compared to Nrf2, Keap1 expression in wild-type yeast was only mildly toxic, but this toxicity was exacerbated by the deletion of the antioxidant genes *BTN1*, *SOD1*, and *TSA2* (Figure 2C). For the full panel of growth assays, refer to Appendix A; for the growth assay control plates, refer to Appendix A. Yeast cells expressing YFP-tagged wild-type Nrf2 or the mutant variants of Nrf2, L30F and T80R, were treated with 300 µM hydrogen peroxide (H_2_O_2_) for 3 h to elicit oxidative stress. A change in Nrf2 localization patterns was observed, as Nrf2-YFP was no longer diffusely spread throughout the yeast cytoplasm and nucleus but formed fluorescent foci (Figure 2D). These foci were especially prominent for yeast cells expressing the Nrf2 T80R mutant variant. Moreover, when yeast cells expressing Keap1-YFP were treated with 300 µM H_2_O_2_, Keap1-YFP formed protein inclusions (Figure 2E). The optimal treatment dose and duration were determined by measuring the cell viability in non-transfected HeLa cells to achieve a moderate dose-dependent response to hydrogen peroxide treatment (Appendix A). These data indicate that increased oxidative stress modifies the expression of Keap1 and induces inclusion formation of both Nrf2 and Keap1 in yeast.

#### 3.1.3. Nrf2 Forms Protein Inclusions under High Oxidative Stress Conditions in HeLa Cells

We next assessed whether Nrf2 also forms oxidative stress-induced protein inclusion formation in human cells. Physiological concentrations of hydrogen peroxide in mammalian cells range between 1 and 10 nM, where supraphysiological concentrations greater than 100 nM are denoted as oxidative stress [48]. To assess what happens to Nrf2 under such high oxidative stress conditions (such as in cancer, where ROS levels are high due to high metabolic activity and genetic instability [49]), HeLa cells were treated with high concentrations of hydrogen peroxide, with the dose and duration determined experimentally as previously mentioned (Appendix A). Figure 3A documents the intramolecular localization of wild-type Nrf2 and two Nrf2 mutant variants with reduced affinity to Keap1 (L30F and T80R) in HeLa cells. HeLa cells were transfected with wild-type or mutant green fluorescent protein (GFP)-tagged Nrf2 and treated with 100 or 300 µM H_2_O_2_ for 3 h. As a negative control, cells were transfected with a pcDNA3.1-GFP empty vector for mammalian expression. Fluorescence microscopy revealed the formation of cytosolic and possibly nuclear protein inclusions of wild-type and mutant Nrf2 in both untreated and treated cells, at endogenous expression levels and even more so when Nrf2 L30F and Nrf2 T80R were overexpressed by transient transfection. In comparison, no stress-induced protein inclusions were observed for control HeLa cells transfected with GFP alone. The percentage of cells with inclusions increased in a hydrogen peroxide dose-dependent manner (Figure 3B). Nrf2 T80R shows a significantly higher percentage of cells with inclusions compared to wild-type Nrf2. Confocal microscopy was used to visualize wild-type Nrf2 inclusions at a higher resolution and Z-stacking revealed that Nrf2 inclusions are situated around the nucleus rather than within it (Figure 3C), thus indicating that these inclusions are cytosolic.

#### 3.1.4. Nrf2 Protein Inclusions Are Not Artifacts, Are Preventable by Certain Antioxidants, and May Remain Functional

To determine whether oxidative-stress induced Nrf2 inclusion formation is an artifact of overexpression by transient transfection, un-transfected HeLa cells were treated with 300 µM H_2_O_2_ for 3 h and observed by immunofluorescence for endogenous Nrf2. The localization patterns for endogenous Nrf2 were similar to that of transfected Nrf2 with hydrogen peroxide treatment (Figure 4A), confirming that this observed effect is likely, not due to Nrf2 overexpression. To determine if Nrf2 inclusion formation is oxidative stress-specific, HeLa cells expressing Nrf2-GFP were treated with 50 µM MG132, a proteasome inhibitor that elicits general protein misfolding stress. Compared to cells treated with 300 µM H_2_O_2_, which formed some Nrf2 inclusions, treatment with MG132 did not result in the formation of cytosolic Nrf2 inclusions (Figure 4B), indicating that inclusion formation is oxidative stress-specific. Furthermore, we explored if treatment with antioxidants, such as vitamin C (ascorbic acid) and N-acetylcysteine (NAC), could prevent Nrf2 inclusion formation. As an antioxidant, vitamin C acts as an ROS scavenger [50] and has been shown to activate the Nrf2 pathway in both cell and animal models [51,52,53]. Similarly, NAC is a thiol-containing antioxidant that can scavenge ROS but can also regulate the redox state of numerous antioxidant proteins (e.g., glutathione (GSH) and thioredoxin (Trx1)) and promote their reduced form [54,55]. Furthermore, numerous studies in cells, rodents, and humans have shown that treatment with NAC promotes Nrf2 activation [56,57,58,59,60]. Thus, given the involvement of these antioxidants in ROS scavenging, redox regulation, and/or Nrf2 activation, we presume that pre-treatment with vitamin C or NAC prevents or minimizes oxidative stress-induced protein misfolding through direct H_2_O_2_ scavenging and/or through the enhanced activity of antioxidant enzymes. To test this, transfected cells were pre-treated with 100 µM vitamin C or 3 µM NAC for 24 h and subsequently treated with 300 µM H_2_O_2_ for 3 h. A reduction in Nrf2 inclusion formation was observed for pre-treatment with NAC (Figure 4C, bottom) but not vitamin C (Figure 4C, top). Finally, we assessed the induction of Nrf2 (*NFE2L2*), Nrf2 target genes (*HMOX1*, *NQO1*), and thioredoxin (*TXN*) in HeLa cells following treatment with 300 µM H_2_O_2_ for 3 h (Figure 4D). Data were normalized to *RPLP0* transcript levels. The Nrf2 target gene *HMOX1* was induced following hydrogen peroxide treatment but *NQO1* was not, presumably because the treatment duration of 3 h was insufficient for induction [61]. Nonetheless, this suggests Nrf2 may retain its function as a transcription factor despite inclusion formation.

### 3.2. Analysis of Keap1

#### 3.2.1. Keap1 Forms Protein Inclusions under High Oxidative Stress Conditions in HeLa Cells

HeLa cells were transfected with GFP-tagged Keap1 and treated with 100 or 300 µM H_2_O_2_ for 3 h. Fluorescence microscopy revealed the formation of cytosolic and possibly nuclear protein inclusions in both untreated and treated cells (Figure 5A). Quantification of the percentage of cells containing inclusions revealed that Keap1 inclusions formed in a hydrogen peroxide dose-dependent manner (Figure 5B). Confocal microscopy was used to visualize these inclusions at a higher resolution (Figure 5C) and Z-stacking revealed that Keap1 inclusions were situated around the nucleus rather than within it (Figure 5D), thus indicating that these inclusions were cytosolic.

#### 3.2.2. Keap1 Protein Inclusions Are Not Artifacts and Are Not Preventable by Pretreatment with Certain Antioxidants

Again, to determine whether oxidative stress-induced Keap1 inclusion formation was an artifact of overexpression by transient transfection, un-transfected HeLa cells were treated with 300 µM H_2_O_2_ for 3 h and observed using immunofluorescence microscopy for endogenous Keap1. Endogenous Keap1 also formed inclusions upon treatment with hydrogen peroxide (Figure 6A), confirming that this observed effect is not due to Keap1 overexpression. To determine if Keap1 inclusion formation is oxidative stress-specific, HeLa cells expressing Keap1-GFP were treated with 50 µM MG132 to elicit general protein misfolding stress. Compared to cells treated with 300 µM H_2_O_2_, which formed Keap1 inclusions, treatment with MG132 did not result in the formation of inclusions (Figure 6B).

Finally, we determined if vitamin C and NAC could prevent Keap1 inclusion formation upon oxidative stress. Transfected cells were pre-treated with 100 µM vitamin C or 3 µM NAC for 24 h and subsequently treated with 300 µM H_2_O_2_ for 3 h. Unexpectedly, no significant reduction in Keap1 inclusions was observed (Figure 6C), which contrasts with our results with Nrf2, where Nrf2 inclusion formation could be prevented by NAC pre-treatment.

#### 3.2.3. Keap1 Forms Oxidative Stress-Induced Protein Inclusions in Breast Cancer Cell Lines

To ensure that Keap1 stress-induced inclusion formation was not a HeLa cell line-specific phenomenon, we treated two human breast cancer cell lines, 21MT-1 and SKBR3, with 300 µM H_2_O_2_ for 3 h and performed immunofluorescence microscopy for endogenous Keap1 expression. Upon high oxidative stress, both cell lines showed Keap1 inclusions (Figure 7A,B), which was quantified to be statistically significantly different from the untreated control (Figure 7C).

### 3.3. Analysis of Nrf2 and Keap1 Purified Protein

#### Purified Proteins for Nrf2 and Keap1 Aggregate upon Exposure to Oxidative Stress

Purified proteins were used to biochemically assess Nrf2 and Keap1 misfolding and inclusion formation (or aggregation) using two methods: (1) traditional SDS-PAGE with Coomassie blue gel staining; and (2) semi-denaturing detergent agarose gel electrophoresis (SDD-AGE) with western bot analysis, a method optimized for detecting protein aggregates [62]. As shown in Figure 8A, purified full-length Nrf2 was incubated with 600 μM H_2_O_2_ in the absence and presence of reducing agent β-mercaptoethanol (βME). Upon treatment with hydrogen peroxide, Nrf2 formed a dense high molecular weight smear, indicating the formation of aggregated higher molecular weight species. With the addition of 5% of the reducing reagent βME, this structure collapsed, indicating that this high molecular weight protein species is, to some degree, dependent on disulfide bonds. Furthermore, analysis of the purified Neh5 domain of Nrf2, which harbors one of the six cysteines in the protein, revealed the formation of an aggregated higher molecular weight species with hydrogen peroxide treatment that also collapsed upon βME treatment (Figure 8B).

Similarly, purified full-length Keap1 protein was treated with 600 μM H_2_O_2_ in the presence and absence of βME. Upon treatment with hydrogen peroxide, Keap1 formed a dense high molecular weight smear, indicating the formation of an aggregated higher molecular weight species. With the addition of 5% βME, this structure collapsed (Figure 8C), indicating that this high molecular weight protein species was also, at least in part, dependent on the formation of disulfide bonds. Purified protein for Keap1′s Kelch domain was then used to determine if this well-folded domain alone, which contains only eight cysteine residues, would misfold under oxidative stress conditions. Unlike full-length Keap1, treatment of the Kelch domain with 600 µM H_2_O_2_ did not produce a high molecular weight species (Figure 8D).

Finally, fractionation assays were used to determine the soluble and aggregated fractions for purified full-length Nrf2 and Keap1 following hydrogen peroxide treatment (Figure 8E). The total purified protein sample was centrifuged and divided into the soluble supernatant fraction and the aggregated pellet fraction and resolved using traditional SDS-PAGE with Coomassie blue gel staining. Both Nrf2 and Keap1 contained soluble and aggregated protein fractions in the supernatant and pellet, respectively, indicating that some protein within the sample remained un-aggregated following treatment.

## 4. Discussion

In this work, we demonstrate that two key proteins of the antioxidant pathway, Nrf2 and Keap1, form intracellular inclusions upon exposure to supraphysiologic levels of oxidative stress. We believe that, at least in part, the aberrant formation of disulfide bonds causes the misfolding and inclusion formation of both proteins, which may either be an adaptive regulatory response or a maladaptive event that impairs protein function. Nrf2′s intrinsically disordered nature [21] may also contribute to its propensity to misfold and form inclusions.

We observe that in both yeast and mammalian cells (HeLa and breast cancer cell lines), the treatment of cells expressing Nrf2 with hydrogen peroxide results in the formation of protein inclusions in a dose-dependent manner. Interestingly, protein inclusion formation was exacerbated for the Nrf2 mutant variants, L30F and T80R, where Keap1-binding to Nrf2′s Neh2 domain at the low- and high-affinity motifs, respectively, is impaired, resulting in a loss of normal Keap1-mediated degradation [13,14]. Oxidatively misfolded Nrf2 accumulates in the cytosol as inclusions, most notably for the Nrf2 T80R mutant, which escapes all Keap1-mediated degradation. This could explain the significantly high levels of Nrf2 T08R inclusions. On the other hand, Nrf2 L30F has impaired binding at the low-affinity motif but may still retain some Nrf2-binding via the high-affinity motif; however, ubiquitination of Nrf2 is unlikely without an intact low-affinity binding site [15]. Future work will further investigate the functional consequences of these mutants and how impaired Keap1-dependent Nrf2 regulation affects oxidative stress-induced inclusion formation.

Numerous domains of Nrf2 have previously been characterized as intrinsically disordered [14,21] and our data support this. While this structural flexibility could allow Nrf2 to bind to a large number of different proteins [63], it may also render Nrf2 susceptible to aberrant protein misfolding. Intrinsically disordered proteins tend to misfold under certain conditions [64], which appears to also be the case for Nrf2 during high levels of oxidative stress. This misfolding and inclusion formation of Nrf2 could be an adaptive mechanism of Nrf2 regulation, i.e., the ‘functional misfolding’ of IDPs [64], or a maladaptive mechanism that impairs protein function. To determine this, further work must investigate the functional consequences of stress-induced inclusion formation of Nrf2 and Keap1. Perhaps under high levels of oxidative stress, Keap1, upon forming aberrant intra- and inter-molecular disulfide bonds, misfolds, is inactivated, and cannot bind to Nrf2 to target it for degradation, thus permanently allowing free Nrf2 to activate the antioxidant response. Conversely, oxidized Nrf2 may be unable to bind Keap1, therefore escaping Keap1-mediated degradation. This again could be an adaptive or maladaptive mechanism of Nrf2 regulation under supraphysiologic oxidative stress conditions (e.g., in cancer). In this regard, determining the reversibility of this cysteine oxidation and protein misfolding event is important, as irreversible oxidative damage and inactivation may lead to constitutive Nrf2 activation, as observed in many human cancers [65]. Accordingly, in rapidly dividing cancer cells where ROS production is high [66], or in the process of ageing where there are marked decreases in antioxidant capacity [67], Nrf2, Keap1, and other proteins may be over-oxidized, resulting in dysregulated, and possibly enhanced, antioxidant activity.

The activity of some proteins is greatly dependent on cysteine residues, which often occur in the functional site of proteins (regulatory, catalytic, cofactor-binding, etc.) [68]. Within the Keap1-Nrf2 antioxidant pathway, specific cysteines within Keap1 are required for sensing oxidative stress and are thus critical for regulation of the antioxidant response [7,8,9,43]. Wild-type Nrf2 contains only six cysteine residues, but our in vitro experiments document that at least some of the cysteines in Nrf2 are oxidized and are central to the formation of high molecular weight protein species. It is plausible that oxidation-induced inclusion formation modifies the transcriptional activity of Nrf2, as He et al. found that some cysteine residues of Nrf2 play important roles in oxidant sensing, Keap1-dependent ubiquitination and degradation of Nrf2, and in the transcriptional activation of ARE-containing Nrf2 target genes [69]. In contrast, Keap1 is not intrinsically disordered and its misfolding may be mostly dependent on the presence of cysteine residues that are highly susceptible to oxidation. Keap1 contains 27 cysteines, 24 of which were found to be highly or completely conserved. Interestingly, except for the chicken and zebrafish, all key sensor cysteines within Keap1 [7,8,9,43] were mostly completely conserved, which demonstrates their importance for Keap1 function. Cysteines are one of the least abundant amino acids in mammals, comprising an average of 2.3% for the human proteome [45]. Yet, we found that human Keap1 contains 4.33% cysteine content, almost double the average for the human proteome [45]. This high content of cysteine residues in Keap1 may render it more susceptible to oxidation, for example, by aberrant disulfide bond formation. Fourquet et al. have shown that hydrogen peroxide can oxidize Keap1 [70].

Apart from the ER, where temporary disulfide bonds help to stabilize proteins during the folding process [71], most cysteines in the cell are kept reduced. Thioredoxin is a ubiquitous antioxidant enzyme chiefly responsible for thiol-redox control to reduce disulfide bonds and protect proteins from oxidative inclusion formation and inactivation [72]. The question remains then, if oxidized Nrf2 and Keap1 inclusions are reversible by thioredoxin-mediated disulfide bond reduction and protein refolding by molecular chaperones, and to what extent are these inclusions tolerated before it becomes irreversible and toxic? This question is particularly important in the context of neurodegenerative diseases where oxidative damage and protein misfolding and aggregation/inclusion formation are common hallmarks across numerous neurodegenerative diseases [73]. Pre-treatment of HeLa cells with the antioxidant NAC (a ROS scavenger and regulator of the intracellular redox state) but not vitamin C (an ROS scavenger only) prevented the formation of Nrf2 inclusions upon oxidative stress. This suggests that while ROS scavenging may help alleviate oxidative cysteine modifications, ROS scavenging alone is insufficient in preventing inclusion formation without the action of antioxidant enzymes, such as thioredoxin. The use of live-cell imaging would be very informative and allow us to observe, using time-lapse microscopy, the formation of these stress-induced inclusions, and to determine if these inclusions can be reversed over time either by endogenous cellular mechanisms or through exogenous chemical treatments. The other arising question is whether molecular chaperones, which assist in the refolding of misfolded proteins, can refold cytosolic, disulfide bond-mediated, misfolded protein inclusions, such as those observed for Nrf2 and Keap1. Within yeast, Hsp104 cooperates with Hsp70 and Hsp40 to function as a disaggregase that mediates the dissolution of protein aggregates and inclusions to restore their function or facilitate their clearance from the cell [74,75]. However, metazoans, including humans, do not possess a known Hsp104 homologue, so it remains unclear how some protein aggregates and inclusions are handled in these cells, and which molecular chaperones are important for this [76].

Under normal conditions, Keap1 is constantly shuttling between the cytosol and the nucleus via importin α7 (also known as karyopherin α6, KPNA6), and the nuclear import of Keap1 represses the Nrf2 antioxidant response [77]; however, our results show that misfolded Keap1 inclusions are cytosolic. Misfolded Keap1 may render the protein inactive, impairing Keap1-mediated degradation of both cytosolic and nuclear Nrf2. We speculate that Keap1, upon forming aberrant intra- and/or inter-molecular disulfide bonds, cannot bind to Nrf2 to target it for degradation, thus allowing free Nrf2 to activate the antioxidant response. It is plausible that this Keap1 inactivation is irreversible and that even newly synthesized Keap1 cannot escape oxidation-based inactivation under high levels of oxidative stress, thus causing a long-lasting constitutive activation of Nrf2. This mechanism may be predominant in cancer, where ROS levels can be high due to increased metabolic activity and genetic instability [49], thereby creating a hostile oxidative stress environment. Importantly, under the high oxidative stress conditions that we explored (i.e., 300 µM H_2_O_2_ for 3 h), misfolded Nrf2 appeared to remain functional and could induce the Nrf2 target gene *HMOX1*; however, the Nrf2 target gene *NQO1* was not induced, likely due to an insufficient treatment duration for this particular gene [61].

Finally, we provide additional evidence that cysteine oxidation of Nrf2 and Keap1 may contribute to the formation of protein inclusions or aggregates upon treatment with hydrogen peroxide using purified proteins, and that treatment with a reducing agent to break disulfide bonds resulted in the solubilization of the aggregated species. Of note, our in vitro data argue that the cysteines within the stably folded Kelch domain of Keap1 are not susceptible to oxidation and misfolding for the Kelch domain alone, which is consistent with the notion that not all cysteine residues within Keap1 are equally reactive. Future work must determine which cysteines in Keap1 are most susceptible to oxidation and inclusion formation (e.g., by cysteine mutation analyses) and if the full-length protein is required for this misfolding to occur.

It is important to mention the work by Taguchi et al., which proposes that oxidative stress causes Keap1 misfolding and its sequestration into inclusion bodies that are removed by p62/SQSTM1 [78]. This is a mechanism that is not mutually exclusive to the one proposed in this study; however, it is also important to note that the inclusion bodies observed by Taguchi et al. seem to differ in morphology compared to the inclusions observed in this study and that the end product of the p62–Keap1 interaction is Keap1 degradation by autophagy, which was not observed here. In addition, purified Keap1 protein misfolds and forms inclusions upon oxidative stress treatment in a biochemical ‘test tube’ scenario where p62 is absent.

Taken together, we employed yeast models, cultured mammalian cells, and purified proteins to assess the oxidative stress-induced inclusion formation of Nrf2 and Keap1. We argue that the intrinsically disordered nature of Nrf2 exposes its cysteine residues to ROS and thus makes it more prone to misfolding while the more structured Keap1′s unusually high content of cysteine residues makes the protein more susceptible to misfolding by aberrant disulfide bond formation. Whether this mechanism is involved in physiologic Nrf2 regulation during high levels of oxidative stress, or if it is a pathological mechanism in hostile stress conditions remains to be explored. Our work provides new and preliminary insight into previously unexplored aspects of Nrf2 regulation by oxidation-dependent protein inclusion formation. Future work will seek to explore the functional outcome of this oxidation event in normal cells and cancer cells.

## Figures and Tables

**Figure 1 antioxidants-11-00243-f001:**
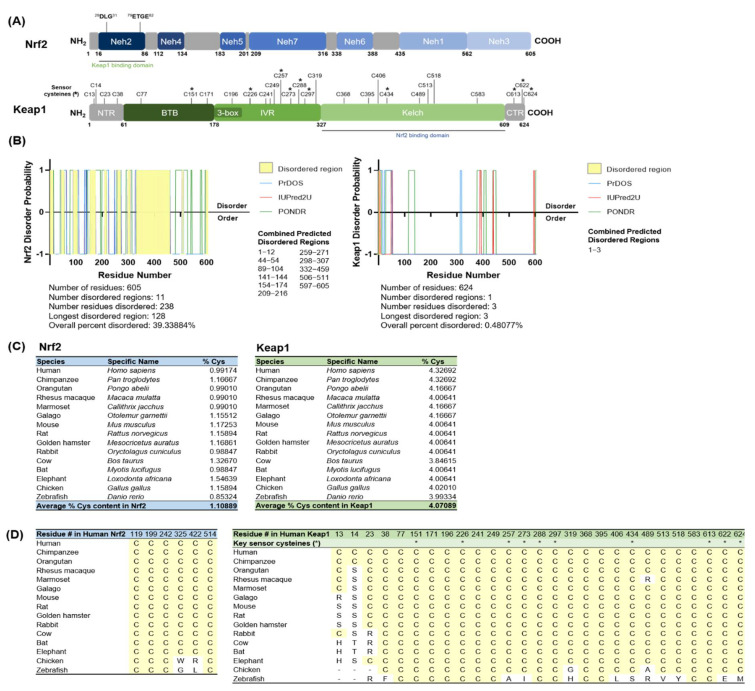
Protein disorder analyses and cysteine analyses for Nrf2 and Keap1. (**A**) Domain maps for Nrf2 and Keap1 show the location of all cysteine residues. Key oxidative stress-sensing cysteines in Keap1 are marked with an asterisk (*). (**B**) Disordered profile plots for Nrf2 and Keap1 predicting the location of intrinsically disordered regions within each protein as predicted by three independent algorithms (PrDOS, IUPred2A, and PONDR). The intrinsically disordered regions predicted by all three algorithms are highlighted in yellow. (**C**) The percentage of cysteine content is calculated for 15 species from human to zebrafish. (**D**) Protein sequence alignment for cysteine residues in Nrf2 and Keap1 across 15 species. All cysteines are highlighted in yellow. Sensor cysteines within Keap1 are marked with an asterisk (*).

**Figure 2 antioxidants-11-00243-f002:**
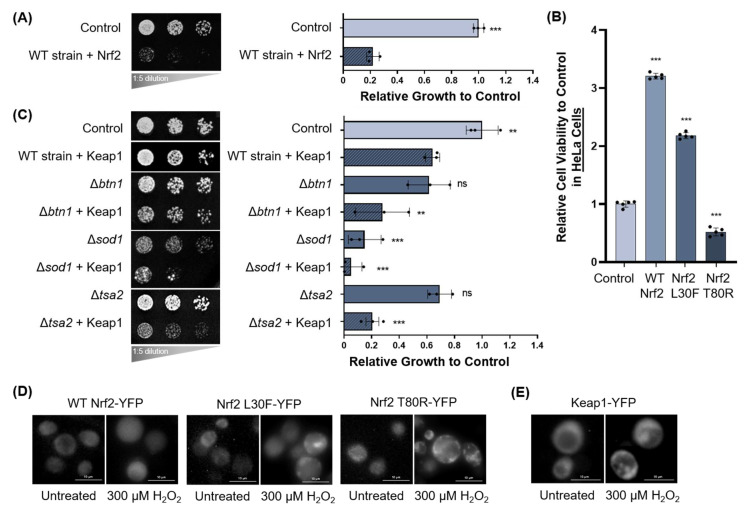
Nrf2 and Keap1 expression is modified by oxidative stress in yeast. (**A**) Growth assays for human Nrf2 transformed into wild-type yeast grown on agar plates. Growth is quantified to the right relative to the empty vector control. Only significant data is shown. (**B**) Relative cell viability of human HeLa cells transfected with wild-type Nrf2 and its mutant variants. (**C**) Human Keap1 transformed into yeast deletion strains for various oxidative stress genes grown on agar plates. Only significant data is shown. Growth is quantified to the right relative to the empty vector control. (**A**–**C**) Means derived from three biological replicates were used during analysis. Means were analyzed using one-way ANOVA followed by Tukey’s post hoc test. Data are expressed as mean ± SD. *p* < 0.05 was considered statistically significant; ** *p* < 0.01, *** *p* < 0.001; *p* = ns (non-significant). (**D**) Yeast expressing YFP-tagged Nrf2 and two Nrf2 mutants treated with 300 µM H_2_O_2_ for 3 h. (**E**) Yeast expressing Keap1-YFP treated with 300 µM H_2_O_2_ for 3 h.

**Figure 3 antioxidants-11-00243-f003:**
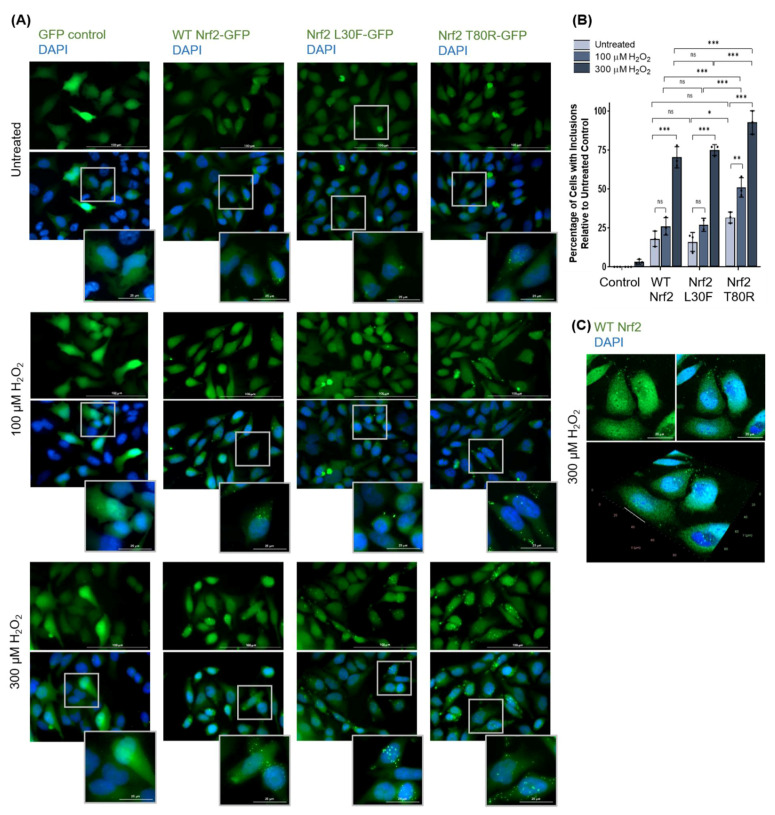
Nrf2 forms inclusions upon exposure to high levels of oxidative stress in HeLa cells. (**A**) HeLa cells transfected with GFP-tagged Nrf2 and two Nrf2 mutants and treated with 100 or 300 µM H_2_O_2_ for 3 h, visualized by fluorescence microscopy. As a negative control, cells were transfected with a pcDNA3.1-GFP empty vector for mammalian expression (note: the GFP channel for the control condition was underexposed to reduce overexposure of very high-expression cells). (**B**) Quantification of Nrf2-expressing cells with inclusions following hydrogen peroxide treatment as observed in (**A**). Means were analyzed using one-way ANOVA followed by Tukey’s post hoc test. Data are expressed as mean ± SD. *p* < 0.05 was considered statistically significant; * *p* < 0.05, ** *p* < 0.01, *** *p* < 0.001; *p* = ns (non-significant). (**C**) Confocal microscopy with Z-stacking for wild-type Nrf2-expressing cells treated with hydrogen peroxide, visualized by immunofluorescence, and demonstrating that Nrf2 inclusions are cytosolic.

**Figure 4 antioxidants-11-00243-f004:**
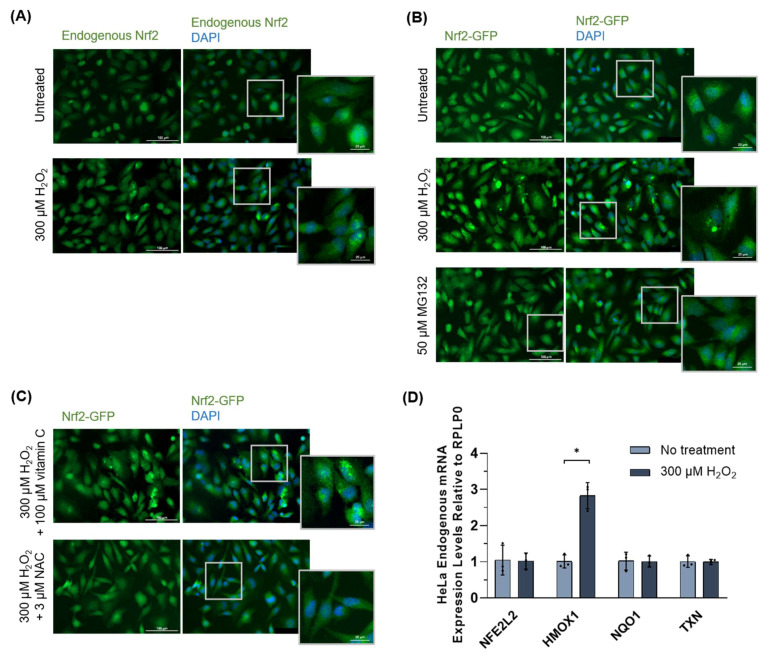
Further analyses of the oxidative stress-induced protein misfolding of Nrf2. (**A**) Endogenous Nrf2 in un-transfected HeLa cells treated with 300 µM H_2_O_2_ for 3 h, visualized by immunofluorescence microscopy. (**B**) Transfected Nrf2-GFP in HeLa cells treated with 50 μM MG132 for 6 h, visualized by fluorescence microscopy. (**C**) Transfected Keap1-GFP in HeLa cells pretreated with 100 μM vitamin C or 3 μM NAC for 24 h followed by treatment with 300 µM H_2_O_2_ for 3 h, visualized by fluorescence microscopy. (**D**) Relative mRNA levels of Nrf2 (*NFE2L2*), Nrf2 target genes (*HMOX1*, *NQO1*), and thioredoxin (*TXN*) were evaluated by RT-qPCR following treatment of HeLa cells with 300 µM H_2_O_2_ for 3 h. Data were normalized to *RPLP0* transcript levels. Means derived from three replicates were used during analysis. Data are expressed as mean ± SD. *p* < 0.05 was considered statistically significant; * *p* < 0.05.

**Figure 5 antioxidants-11-00243-f005:**
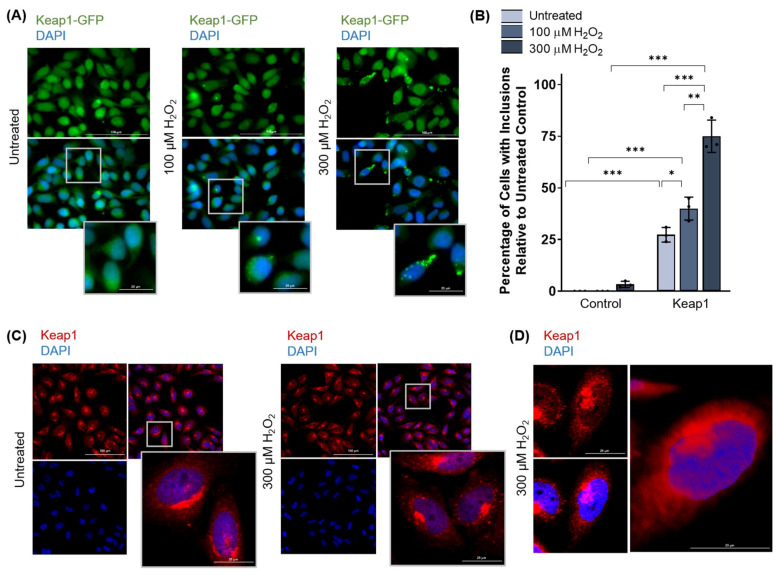
Keap1 forms inclusions upon exposure to high levels of oxidative stress in HeLa cells. (**A**) HeLa cells transfected with Keap1-GFP and treated with 100 or 300 µM H_2_O_2_ for 3 h, visualized by fluorescence microscopy. (**B**) Quantification of Keap1-expressing cells with inclusions following hydrogen peroxide treatment in (**A**). Means derived from three biological replicates were used during the analysis. Means were analyzed using one-way ANOVA followed by Tukey’s post hoc test. Data are expressed as mean ± SD. *p* < 0.05 was considered statistically significant; * *p* < 0.05, ** *p* < 0.01, *** *p* < 0.001. (**C**) Confocal microscopy for Keap1-expressing cells treated with 300 µM H_2_O_2_ for 3 h, visualized by immunofluorescence microscopy. (**D**) Confocal microscopy with Z-stacking for Keap1-expressing cells treated with hydrogen peroxide, visualized by immunofluorescence, and demonstrating that Keap1 inclusions are cytosolic.

**Figure 6 antioxidants-11-00243-f006:**
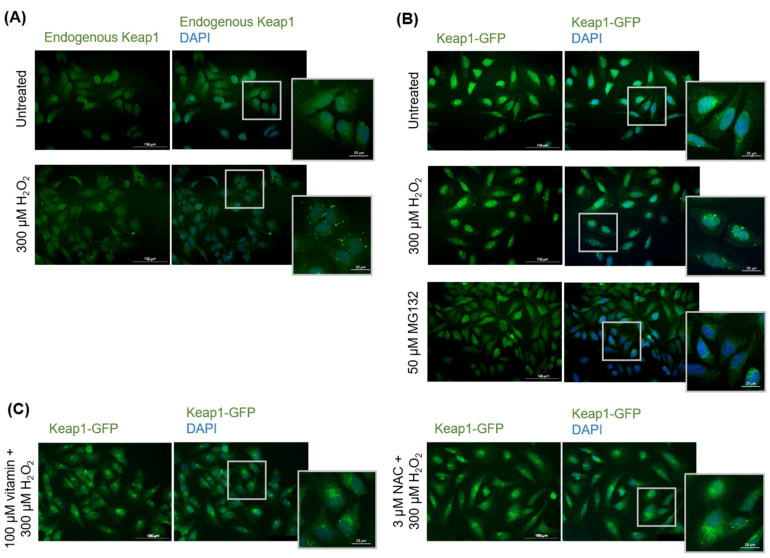
Further analyses of Keap1 oxidative stress-induced protein misfolding. (**A**) Endogenous Keap1 in un-transfected HeLa cells treated with 300 µM H_2_O_2_ for 3 h, visualized by immunofluorescence microscopy. (**B**) Transfected Keap1-GFP in HeLa cells treated with 50 μM MG132 for 6 h, visualized by fluorescence microscopy. (**C**) Transfected Keap1-GFP in HeLa cells pretreated with 100 μM vitamin C or 3 μM NAC for 24 h followed by treatment with 300 µM H_2_O_2_ for 3 h, visualized by fluorescence microscopy.

**Figure 7 antioxidants-11-00243-f007:**
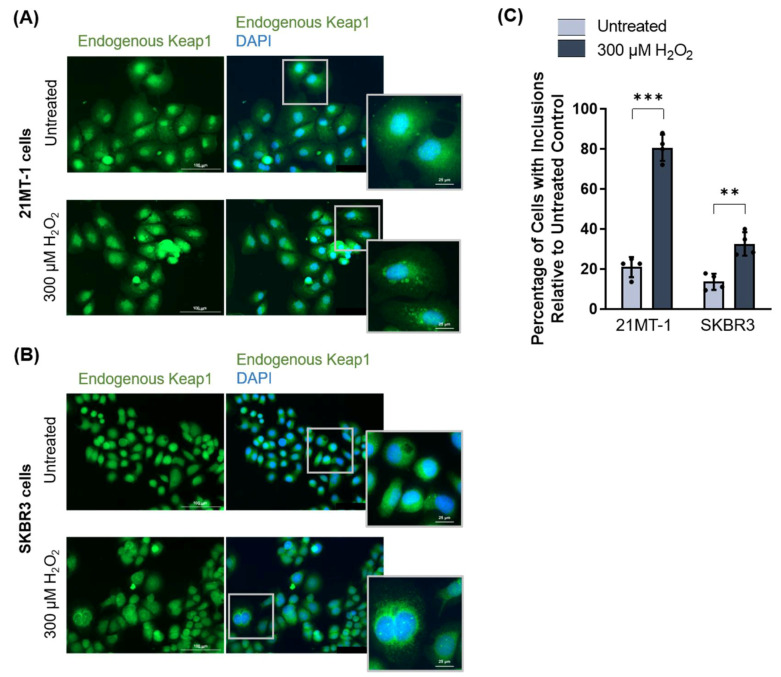
Keap1 forms oxidative stress-induced inclusions in breast cancer cell lines. (**A**,**B**) Endogenous Keap1 expression in two breast cancer cell lines, 21MT-1 (A) and SKBR3 (B), treated with 300 µM H_2_O_2_ for 3 h, visualized by immunofluorescence microscopy. (**C**) Quantification of Keap1 inclusions following hydrogen peroxide treatment in (**A**,**B**). Means derived from three biological replicates were used during the analysis. Means were analyzed using one-way ANOVA followed by Tukey’s post hoc test. Data are expressed as mean ± SD. *p* < 0.05 was considered statistically significant; ** *p* < 0.01, *** *p* < 0.001.

**Figure 8 antioxidants-11-00243-f008:**
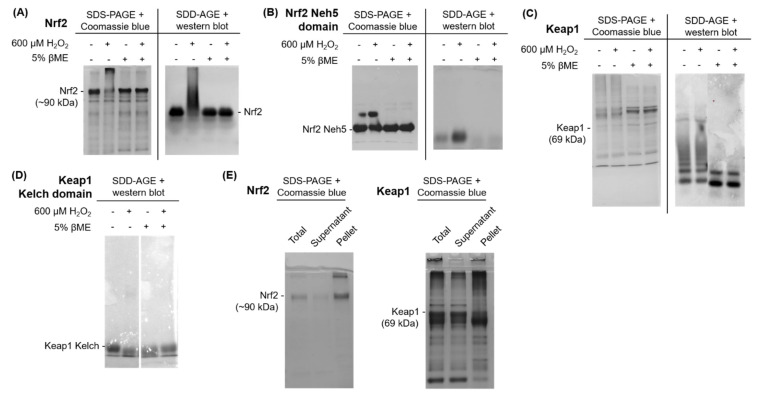
Nrf2 and Keap1 purified proteins aggregate upon exposure to oxidative stress in vitro. Purified protein treated with 600 μM H_2_O_2_ ± βME for 30 min for (**A**) full-length Nrf2, (**B**) Nrf2′s Neh5 domain, (**C**) full-length Keap1, and (**D**) Keap1′s Kelch domain. (**E**) Fractionation assay for purified full-length Nrf2 and Keap1 protein. All purified proteins were resolved using SDS-PAGE with Coomassie Blue gel staining and/or by SDD-AGE with western blot detection, as indicated.

**Table 1 antioxidants-11-00243-t001:** Primer sequences utilized for RT-qPCR.

mRNA Probe	NCBI Gene Accession Number	Primer Sequences (Forward and Reverse, 5′ to 3′)
NFE2L2	AC079305	F: GCCCAATGTGAGAACACACC R: TGTGAGATGAGCCTCCAAGC
HMOX1	AY460337	F: CCCCAACGAAAAGCACATCC R: AGACAGCTGCCACATTAGGG
NQO1	AH005427	F: TGGAAGAAACGCCTGGAGAAT R: CTGGTTGTCAGTTGGGATGG
TXN	AF548001	F: ATTGTGACCAGCACCTACGG R: CATGGTGGAGTTGTCCCGAA
RPLP0	AC004263	F: CCTCATATCCGGGGGAATGTG R: GCAGCAGCTGGCACCTTATTG

## Data Availability

The data presented in this study are available in article and Appendix A.

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
