# Peer review of "Oxidative Stress-Induced Misfolding and Inclusion Formation of Nrf2 and Keap1"

_antioxidants, 2022, doi:10.3390/antiox11020243_

Round 1

Reviewer 1 Report

The paper “Oxidative stress-induced misfolding and inclusion formation 2 of Nrf2 and Keap1” is very interesting and gives a novel perspective about the Keap1/Nrf2 activation and response to oxidative insults. Given the central role of Nrf2 in different diseases, the understanding of its regulation is very timely and relevant.

However, I think the work needs some additional assays to prove the Authors’ point of view.

I my opinion this work needs clarification if the inclusions/aggregation of Nrf2 shown in detail are indicative of damaged Nrf2 as Authors claim (Line 429 - “Oxidatively damaged Nrf2 accumulates in the cytosol as inclusions,”), or just the physiologic response to oxidative stress delivering a functional Nrf2. The clarification of this would account for the possible modulation of these aggregates as therapeutic targets.

Some major points should be addressed to complete the study. Minor points are also indicated to improve the manuscript.

Major points:

1 - The biological consequence of Nrf2 aggregation should be more explored, and a mechanistic explanation is needed. Is Nrf2 functional? Does it bind to DNA in ARE sequences? In longer incubation periods are other downstream targets of Nrf2 also expressed? HO-1 is very “responsive it is OK as a control, but other antioxidant enzymes should be addressed.

2 - Although good anti-Nrf2 antibodies are scarce, some of the assays could be repeated only to prove that inclusions are not artifacts.  

3 - Figure 8C – image presented is not convincing for Keap 1 aggregation. No higher molecular weight bands or smear can be observed after treatment with H2O2

4 – Authors do not explain how purified proteins used in figure 8 were obtained – this information should be added

Minor points:

5 - Figure 3A – the GFP-Control images (higher amplification) seem to be unfocused. Do the Authors have alternative figures to show?

6 - In all microscopy photographs a magnification bar should be included.

7 - What was the method used to count the inclusion in the cells? What was the criteria used to define cells having or not inclusions? How many cells were counted?

8 - Abstract:  Nrf2 and Keap1 should be defined in full

9 - Material and Methods

9.1 - 2.5. Cell viability assays: range of concentrations and time points should be indicated

9.2 - 2.6. Fluorescence and immunofluorescence microscopy: were cells treated with H2O2? Concentration and time point?

9.3 - 2.7. RNA isolation and quantitative reverse transcription PCR (RT-qPCR): concentration of H2O2 treatment should be indicated (not just 300 ul).

10 - Results:

10.1 - Line 202 – small typo: “Keap1 contains has a high 4.33%”; delete “has” or “contains”

10.2 - Line 226 – delete one “that”

10.3 - Figure 2 – how long was the treatment with H2O2?

Author Response

Please see theattachment.

Reviewer 2 Report

In the article by Ngo et al. the authors attempt to shine light on the fate of two key proteins of the antioxidant pathway, Nrf2 and Keap1, under oxidative stress. I find the study well planned and executed although the results are a bit counter-intuitive.

Upon oxidative stress Keap1 should be inactivated, while Nrf2 should be stabilized in order to express antioxidant genes.

  1. The authors show that the Nrf2 upon oxidative stress is toxic to the cells. On the other hand, they claim that Nrf2 is still functional despite inclusion formation. Wouldn’t this effect be visible in the growth phenotype (i.e. expression of antioxidants should have helped the cells to survive)?

  1. Under the presence of antioxidants the authors state that they observe a reduction in inclusion formation of Nrf2. Is the reduction of inclusion formation of Nrf2 due to the fact that less Nrf2 is misfolded or is it due to less expression of Nrf2? The total amount of Nrf2 in the cell should be compared with and without the presence of antioxidants.

  1. This is connected to point 2. Under the presence of antioxidants there is no reduction in inclusion formation of Keap1. I would expect actually the opposite to be true. If antioxidants are present Keap1 should not be inactivated/misfolded through oxidation of cysteines. This would also promote more degradation of Nrf2. This could be a reason for the observation in point 2.

  1. I don’t see the smear of Keap1 in the SDS-PAGE in the presence of peroxide in figure 8C. All the bands in the SDS-PAGE seem the same to me (without having any effect) unlike the Nrf2 SDS-PAGE in figure 8A for example. I see a smear only in the SDD-AGE, but also there the smear seems to be present even without the peroxide.

  1. In general to me it seems that especially the Nrf2 is not able to do what it is supposed to be doing, i.e. be stabilized and try to deal with the oxidative stress. On the contrary it seems to be misfolded/aggregated and toxic to the cell. How do the authors explain this “inconsistency” to the existing knowledge?

Minor typos

line 149:  destining à destaining

line 152: agaraose à agarose

line 202: contains has à contains

line 203: which is highly almost double the human average à which is highly almost double than the  human average

line 226: that that à that

line 446-447: We furthermore show that that cysteine oxidation Keap1 may contribute à We furthermore show that that cysteine oxidation of Keap1 may contribute

Round 2

Reviewer 1 Report

The Authors have provided an improved form of the manuscript. They tried to answer most of the comments, and I agree with most of the responses given. However, although this study is well carried on and robust in terms of descriptive data, I still think it lacks a more functional approach. I don’t think that at this point single cell analysis would be the only way to do that, although it would be the best for sure.

For example, the expression of other Nrf2 targets besides HO-1 with longer exposure time points, nuclear translocation and/or DNA binding should be addressed. A global analysis of these experiments together with data regarding cell inclusions would give the idea of Keap1/Nrf2 functionality.

Additionally, cells should be treated with H2O2 and/or NAC or vitamin C antioxidants (for longer periods), general known inducers of Nrf2 expression and activity (curcumin, Sulforaphane, Ursodiol…), and inhibitors of Nrf2 nuclear translocation. This will also rule out the hypothesis that the antioxidant effects of NAC or vitamin C would buffer oxidative species acting upstream Nrf2 activation, and therefore no oxidative stress would trigger the pathway and no inclusions would be generated. In contrast of being a direct effect in cysteine oxidation in Keap1 and/or Nrf2 (a more direct regulation of the pathway).

This work represents a new vision of Nrf2 activation and activity that is a bit counter-intuitive considering what is known about the activity of the Keap1/Nrf2 pathway. In my opinion this novelty should be better consolidated.

Author Response

The Authors have provided an improved form of the manuscript. They tried to answer most of the comments, and I agree with most of the responses given.

We thank the reviewer for acknowledging that the revision improved our manuscript and for agreeing with most of our responses.

However, although this study is well carried on and robust in terms of descriptive data, I still think it lacks a more functional approach. I don’t think that at this point single cell analysis would be the only way to do that, although it would be the best for sure.

We thank the reviewer for recognizing the robustness our data. We disagree, however, that our study is merely descriptive. We find that that Nrf2 and Keap1 form inclusions in vivo and in vitro. We determine the underlying molecular mechanisms as the aberrant oxidation of cysteines, which is, at least partially, reversible. This has never been determined before at this mechanistic level.

For example, the expression of other Nrf2 targets besides HO-1 with longer exposure time points, nuclear translocation and/or DNA binding should be addressed. A global analysis of these experiments together with data regarding cell inclusions would give the idea of Keap1/Nrf2 functionality.

We would like to point out that our microscopic data show that some of the Nrf2 and Keap1 inclusions are indeed localized to the nucleus, whereas other inclusions are found in the cytoplasm.

Extended functional follow-up studies are, in our opinion, way beyond the scope of this study, and will be addressed in future experiments. It will be important to dissect the impact on Nrf2 and Keap1 inclusion formation on their function but also the possible toxic effects of these inclusions, which may very well be independent of their function. Simple transcriptional analyses will certainly not suffice and will most likely be inconclusive.

Additionally, cells should be treated with H2O2 and/or NAC or vitamin C antioxidants (for longer periods), general known inducers of Nrf2 expression and activity (curcumin, Sulforaphane, Ursodiol…), and inhibitors of Nrf2 nuclear translocation. This will also rule out the hypothesis that the antioxidant effects of NAC or vitamin C would buffer oxidative species acting upstream Nrf2 activation, and therefore no oxidative stress would trigger the pathway and no inclusions would be generated. In contrast of being a direct effect in cysteine oxidation in Keap1 and/or Nrf2 (a more direct regulation of the pathway).

Our experiments aimed to determine how Nrf2 and Keap1 inclusions form in cells and in vitro (purified proteins) in physiologically relevant, non-lethal conditions, i.e. treatment with hydrogen peroxide. We agree that studies employing long-term exposure to lower oxidative stress levels could be informative. Yet we also foresee that the cellular response under these conditions will highly depend on the cell type and disease context, e.g. cancer cells versus non-dividing aging neuronal cells. 

This work represents a new vision of Nrf2 activation and activity that is a bit counter-intuitive considering what is known about the activity of the Keap1/Nrf2 pathway. In my opinion this novelty should be better consolidated.

We agree that Nrf2 and Keap1 inclusion formation presents are new and potnetially important finding and there are many plausible functional and possible cytotoxic outcomes. We think these outcomes may highly depend on the cellular/physiological context (e.g. cancer versus aging neuronal tissues) and we hope that our study will trigger many follow-up studies investigating Nrf2 and Keap1 inclusion formation and the functional consequences and pathological roles under different stress conditions, different disease scenarios, and in different experimental models. 

Round 3

Reviewer 1 Report

I still think that this study lacks a more functional approach. Even claiming that different cells may have different responses, additional experiments could have been done testing 2 different kinds of cells with different expected outcomes. This would further sustain the Authors’ main conclusions.

Author Response

I still think that this study lacks a more functional approach. Even claiming that different cells may have different responses, additional experiments could have been done testing 2 different kinds of cells with different expected outcomes. This would further sustain the Authors’ main conclusions.

We thank the reviwer for their comments. Yet, as indicated repeatedly before, we respectfullt disagree as functional studies would entail careful cosideration of specific scenarios (cell type, growth conditions, stress conditions, disease model) as these result in different expression and inudction  levels of Nrf2 and many proteins associated with it. We therefore refer these experiments to future studies.

We would like to emphasize that our major conclusion, i.e. that Nrf2 and Keap1 form inclusions upon oxidation, which we mechansitically decipher, stand with the data we provided in this manuscript.